# Cholangioscopy-Assisted Laser Lithotripsy for Treatment of Postcholecystectomy Mirizzi Syndrome: Case Series

**Bozhidar Hristov** [1,2,*], **Daniel Doykov** [1,2], **Vladimir Andonov** [1,2], **Deyan Radev** [1,2], **Krasimir Kraev** [3,4], **Petar Uchikov** [5,6], **Gancho Kostov** [5,7], **Siyana Valova** [8,9], **Eduard Tilkiyan** [8,9] **and Katya Doykova** [10,11]

1   Second Department of Internal Diseases, Section "Gastroenterology", Medical Faculty, Medical University of Plovdiv, 6000 Plovdiv, Bulgaria
2   Gastroenterology Clinic, University Hospital "Kaspela", 4001 Plovdiv, Bulgaria
3   Department of Propedeutics of Internal Diseases, Medical Faculty, Medical University of Plovdiv, 6000 Plovdiv, Bulgaria
4   Rheumatology Clinic, St. George University Hospital, 6000 Plovdiv, Bulgaria
5   Department of Special Surgery, Faculty of Medicine, Medical University of Plovdiv, 6000 Plovdiv, Bulgaria
6   Second Department of Surgery, St. George University Hospital, 4000 Plovdiv, Bulgaria
7   Department of Surgery, University Hospital "Kaspela", 4001 Plovdiv, Bulgaria
8   Second Department of Internal Diseases, Section "Nephrology", Medical Faculty, Medical University of Plovdiv, 6000 Plovdiv, Bulgaria
9   Clinic of Nephrology, University Hospital "Kaspela", 4001 Plovdiv, Bulgaria
10   Department of Diagnostic Imaging, Medical Faculty, Medical University of Plovdiv, 6000 Plovdiv, Bulgaria
11   Department of Diagnostic Imaging, University Hospital "Kaspela", 4001 Plovdiv, Bulgaria
*   Correspondence: hristov.bozhidar@abv.bg; Tel.: +359-88-4278187

**Abstract:** Introduction. Mirizzi syndrome (MS) represents a rare clinical entity caused by impaction of one or multiple stones in the infundibulum of the gall bladder or the cystic duct resulting in partial or complete obstruction of the common hepatic or common bile duct (CBD). Though described more than a century ago, MS is still one of the most challenging diseases in the spectrum of biliary pathology. In recent years, endoscopic treatment has become an increasingly popular treatment modality. Patients and methods. Three consecutive patients subjected to cholangioscopy-assisted laser lithotripsy (CA-LL) for postcholecystectomy MS (pMS) were retrospectively evaluated. Case reports. Successful clearance of the cystic duct was achieved in all patients in one or two sessions. One complication in the form of mild cholangitis was observed. Clinical success was 100%. Discussion. According to current research, CA-LL achieves a high rate of ductal clearance and acceptable complication rate in patients with pMS. A 250 μm laser fiber seems to be the optimal choice for CA-LL. Our results suggest that procedure duration is closely associated to the stone size and possibly to the operator experience. In our opinion, upon obtainment of successful ductal clearance and drainage, prophylactic stenting does not improve clinical outcome. Conclusions. Our results demonstrate that CA-LL is a safe and effective treatment for pMS.

**Keywords:** Mirizzi syndrome; cholangioscopy; endoscopy; laser lithotripsy; cholecystitis

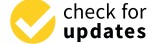



## 1. Introduction

Mirizzi syndrome (MS), also known as extrinsic bile compression syndrome, represents a rare complication of gall stone disease. This clinical entity is induced by impaction of one or multiple stones in the infundibulum of the gall bladder or the cystic duct (CD) resulting in inflammation and partial or complete obstruction of the common hepatic or common bile duct (CBD), with subsequent development of cholecystobiliary (rarely cholecystoenteric) fistula [1–4]. Despite being known and extensively evaluated for decades, MS still represents one of the most challenging complications of gall stone disease and its optimal management is yet to be determined.

Epidemiological studies show that in patients subjected to cholecystectomy or endoscopic retrograde cholangiopancreatography (ERCP), MS is established in 0.7–25% and 1.07% of cases, respectively [5,6]. Preoperative diagnosis remains the determinant factor for successful management (reducing morbidity by 54%) [7], though this is correctly set in merely 8–62.5% of the patients. The gold standard for diagnosis to date remains ERCP with a sensitivity between 76.2% and 100% [8,9]. The chief restraint when considering the procedure is its non-negligible complication rate of around 6–12% (around 4% in patients with MS) [10,11]. Magnetic resonance cholangiopancreatography (MRCP) is considered comparable to ERCP in terms of diagnostic capability. Its overall diagnostic accuracy varies between 69–83%, but this is considerably lower for the establishment of cholecystobiliary fistula (7.4%) [12].

Management of MS largely depends on its type and the time of diagnosis (either pre-, intra- or postoperative). Csendes classification is the most extensively utilized for the stratification of MS [13]. It is based on the presence and extent of cholecystobiliary and cholecystoenteric fistula (Figure 1).

| Type | Description |
|---|---|
| I | Extrinsic compression of the common bile duct by an impacted gallstone |
| II | Cholecystobiliary fistula secondary to an eroded gallstone involving one third of the circumference of the common bile duct |
| III | Cholecystobiliary fistula involving two thirds of the circumference of the common bile duct |
| IV | Cholecystobiliary fistula comprising the whole circumference of the common bile duct |
| V | Any type plus a cholecystoenteric fistula |
| Va | Without gallstone ileus |
| Vb | With gallstone ileus |

**Figure 1.** Csendes classification of Mirizzi syndrome.

A more concise alternative to this classification, widely adopted in clinical practice, is the McSherry classification, which distinguishes two types of MS—1. Type I—CHD compression without fistula; 2. Type II—Presence of cholecystocholedochal fistula [14].

Surgical treatment, particularly open cholecystectomy, remains to date the gold standard for management of MS. Unfortunately, postoperative morbidity remains high (15–20%), with a 6% reoperation rate [7,10]. The laparoscopic approach further increases complications (up to 60%) and is associated with high conversion rate (up to 40%) [7,10,15].

Endoscopic management of MS is an emerging therapeutic modality, which is recently more extensively evaluated as a definitive treatment of MS, with a high success rate of 72–100%, and an acceptable rate of adverse events [16,17]. Endoscopic options for management of MS type I include biliary drainage by means of endoscopic sphincterotomy (ES) and stone extraction with a balloon or Dormia basket, or mechanical lithotripsy for larger stones. If engagement of the stone is impossible, alternative methods are stone fragmentation by means of extracorporeal shockwave lithotripsy (EWSL) or cholangioscopy-assisted (CA) electrohydraulic (EHL)/laser lithotripsy (LL).

Even scarcer is the information on the treatment of postcholecystectomy Mirizzi syndrome (pMS), defined as the establishment of an impacted stone in the cystic duct causing

extrinsic compression in patients already subjected to cholecystectomy. To our knowledge in the current literature there are only two articles (three patients in total) reporting endoscopic treatment of pMS, with only one case of pMS treated by CA-EHL [18,19]. Mainly applied in the field of urology, laser techniques under cholangioscopic guidance are becoming increasingly popular [20–22]. Herein we present our experience with three patients with pMS treated by cholangioscopy-assisted laser lithotripsy (CA-LL).

## 2. Patients and Methods

### 2.1. Patient Selection

Three consecutive patients with pMS treated using CA-LL between January and November 2021 in the Gastroenterology clinic of University hospital "Kaspela" were retrospectively included in the current series. All patients endured laparoscopic cholecystectomy (LC) 1–5 months prior procedure. In all patients, type 1 MS was established prior LC, through ERCP was performed preoperatively due to obstructive jaundice with or without cholangitis, but surgical removal of the retained stones proved to be impossible. In all patients there was at least one unsuccessful attempt for endoscopic removal of the stone using standard techniques (balloon/Dormia basket extraction or mechanical lithotripsy). An oral and written informed consent was obtained prior the procedure, with the patients and their relatives being thoroughly informed on the possible clinical outcomes, adverse events and complications, as well as on the valid alternatives.

### 2.2. Technique Description

In all cases thorough blood tests were performed prior the procedure, including complete blood count, CRP, bilirubin, alkaline phosphatase (AP), gamma-glutamyl transferase (GGT), alanine aminotransferase (ALAT), aspartate aminotransferase (ASAT), amylase, lipase, serum protein, albumin, and electrolytes. Abdominal ultrasonography was performed before the procedure and the findings were thoroughly recorded. All procedures were performed by an endoscopist performing 250 ERCP and 30 cholangioscopic procedures per year.

The CA-LL was executed under general anesthesia using a combination of fentanyl, midazolam, sevoflurane, suxamethonium (Lysthenon), atracurium besylate (Tracrium) and propofol. All patients received prophylactic antibiotics (ceftriaxone 2.0 g i.v. prior the procedure and at least 3 days after). Insufflation with $CO_2$ was utilized instead of ambient air to improve patient's comfort and to mitigate the risk of air embolism.

Patient was placed in left lateral position. A therapeutic duodenoscope Olympus TJF-160VR (Olympus, Hamburg, Germany) was introduced and placed at the second portion of the duodenum "en face" with the major duodenal papilla. In all cases there was a 10 fr plastic stent inserted transpapillary, which was removed with a snare. The common bile duct was then cannulated using a sphincterotome (TrueTomeTM; Boston Scientific, Marlborough, MA, USA) and a 0.035 inch guidewire (JagwireTM (straight type), Boston Scientific, Marlborough, MA, USA). A cholangiogram was obtained to assess ductal anatomy and stone size and number. In all cases, sphincterotomy, which was decided to be sufficient, was performed prior procedure. Continuous flushing with 0.9% NaCl was performed to clear the bile ducts from any contrast media and debris. After that, the sphincterotome was removed and a cholangioscope (SpyGlassTM DS II; Boston Scientific, Marlborough, MA, USA) was introduced along the guidewire in the common bile duct. Using constant irrigation with 0.9% NaCl, the bile tree was distended adequately to visualize the impacted stone (one or multiple) in the cystic duct. A laser fiber 250 μm was introduced then through the working channel of the cholangioscope. Laser lithotripsy of the stone was performed using a Karl Storz Calculase II Holmium laser system (Karl Storz, Tuttlingen, Germany) set at 1.2 J/10 Hz (Figure 2). Laser bursts of <5 s of duration were applied under continuous saline irrigation to fragment the target stone. Balloon extraction of the sludge and stone fragments was intermittently performed in order to maintain optimal visual field. After lithotripsy was completed, a thorough cholangioscopic evaluation of the cystic duct and the bile tree up to the sectoral bile ducts was performed to

verify ductal clearance. Eventually, the remaining fragments were removed with a balloon or Dormia basket and occlusive cholangiogram was obtained to further confirm stone removal. No stent was inserted at the end of the procedure.

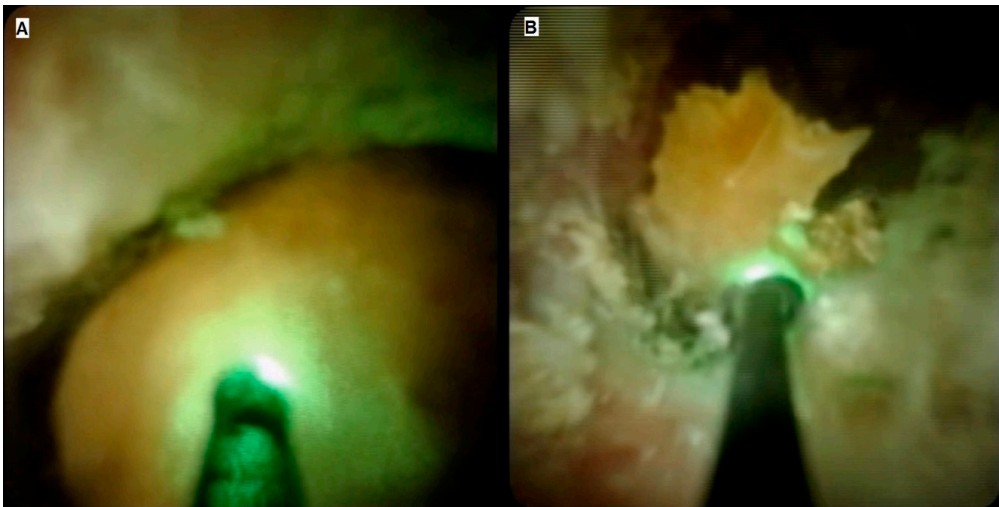

**Figure 2.** Laser lithotripsy: (**A**) At the beginning of the procedure. (**B**) Final result.

All patients underwent ultrasonography and lab testing on post-procedure (PP) day 1 and were discharged on day 2 if no imaging, laboratory or clinical signs of inflammation or biliary obstruction were found. Follow-up ultrasonography was performed one week after the procedure and as per necessity thereafter. MRI was considered as a follow-up method, but was eventually determined to be not cost-effective in clinically fit patients.

Technical success was defined as successful extraction of the stone impacted in the cystic duct, with no residual fragments or strictures requiring stenting to maintain biliary drainage. Clinical success was defined as no signs of obstructive jaundice (icterus, dark urine, pale stools, bilirubin <22.2 mcmol/L, undilated bile ducts on ultrasonography) and cholangitis (fever, chills, ride upper quadrant pain, abnormal liver enzymes) 2 weeks after the procedure.

## 3. Case Reports

### 3.1. Case 1

A 66-year-old woman was transferred from Clinic of Nephrology due to upper right quadrant pain, elevated liver enzymes (ALAT—95.0, GGT—280.0, AP—554.0) and imaging data for dilated bile ducts and gallstone disease established through CT. ERCP was performed, cannulation was achieved with standard guidewire technique. On cholangiography, one 5 mm stone in the common bile duct (CBD) and a large 4 cm stone in the CD protruding to the CBD (MS type I) were established. A sphincterotomy was performed and the CBD stone was removed. Every attempt for entrapment of the CDS failed, so a guidewire was advanced above the stone and a 10 fr by 8 cm plastic stent was inserted to maintain biliary drainage. The patient was then referred for elective surgery. Conventional retrograde laparoscopic cholecystectomy was performed 3 months later, but surgical removal of the stone was impossible due to massive adhesions and high risk for bile duct injury. Since surgical approach was determined to be high risk, the patient was again sent to our department for further endoscopic evaluation. It was determined that conventional extraction of the impacted stone was unfeasible, so CA-LL was discussed. The procedure was electively executed 1 month after the LC according to the described technique. A massive 4 cm cystic duct stone (CDS) was established. Despite considerable reduction of stone size, after 3 h, we failed to fragment the stone sufficiently to achieve complete ductal clearance. Fragments were removed, a new 10 fr/10 cm plastic stent was positioned in the CBD and the patient was scheduled for second LL session. It was performed one month

later and the duration was 1.5 h. Eventually, sufficient fragmentation for extraction of the stone was achieved. Cholangioscopy and cholangiography found no residual stones or strictures with normal evacuation of contrast media from the bile duct. The patient was discharged on PP day 2. In summary, 6 months, a total of four ERCP and two cholangioscopy LL sessions, with a total duration of 4.5 h were needed to achieve CD clearance. No complications associated with ERCP and CA-LL were observed. The patient has been followed-up with for 13 months now, and is symptom free so far.

### 3.2. Case 2

A 44-year-old man was admitted with fever (38.5 °C), epigastric and right upper quadrant pain and jaundice. Lab tests established elevated liver enzymes (ASAT—486, ALAT—840, GGT—399, AP—333), bilirubin (109.1—total; 61.9—conjugated), amylase—498, lipase—344 and CRP—34.8, suggestive of biliary pancreatitis and cholangitis. The leukocyte count was normal. Urgent ERCP was performed, which established a mildly dilated CBD (up to 10 mm) with distal narrowing, a short distal CBD with low insertion of the CD, which was significantly dilated up to 15 mm, with three stones impacted in it (MS type I) (Figure 3). Sphincterotomy was performed in a standard manner up to 10 mm. Every attempt to advance an accessory (including 0.025 inch guidewire) in the CD above the stones was unsuccessful. Bile was aspirated for microbiology and a 10 fr/7 cm plastic stent was placed for biliary drainage and resolution of cholangitis. Recovery was uneventful and patient was scheduled for elective cholecystectomy. An antegrade LC was performed with the CDS supposedly removed, but with no intraoperative cholangiography (IOC). Patient returned 1 month later symptom-free for scheduled removal of the plastic stent. Unfortunately, upon obtainment of cholangiogram we found that the three stones persisted in the bile tree with two of them dislodged in the CBD and one still impacted in the CD. A new 10 fr/7 cm stent was placed. After discussion with the patient, CA-LL during the same admission was planned. It was performed the next day. All three stones were successfully fragmented in single session with consequent extraction of the fragments with a Dormia basket. The whole procedure took 84 min. An episode of chills and fever up to 38.4 °C was seen immediately after the procedure which was successfully managed with Paracetamol 500 mg i.v. No leukocytosis or CRP elevation was found on the following day and the patient was discharged on PP day 2 with a prophylactic antibiotic (Levofloxacine 500 mg/d p.o. for five days). Three ERCP and one CA-LL sessions over two months were needed to achieve CD clearance. The patient has been followed-up with for 17 months now.

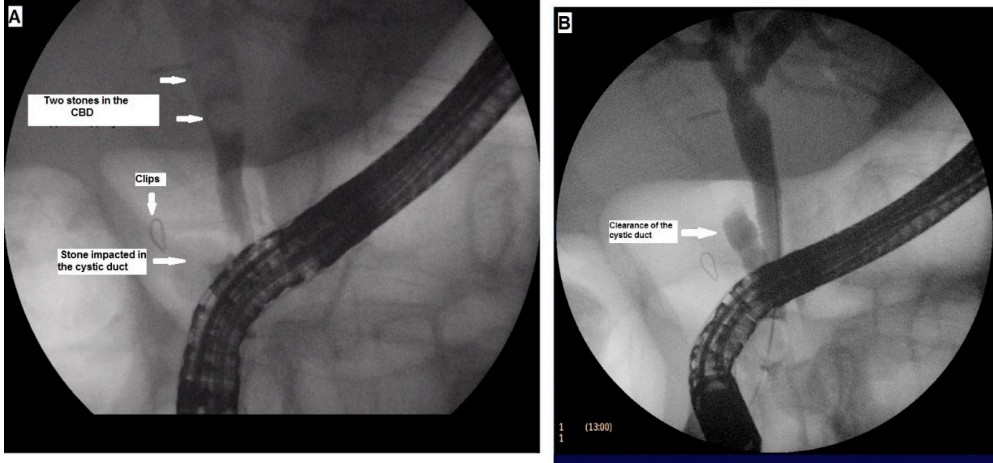

**Figure 3.** Fluoroscopic image. (**A**) At the beginning of the procedure. (**B**) Final result.

### 3.3. Case 3

A 37-year-old woman with cerebral palsy was referred to our department. She was admitted to surgical department in another hospital with right upper quadrant pain,

nausea, vomiting and jaundice. Abdominal US revealed gall stone disease and dilated intra- and extrahepatic bile ducts suggestive of choledocholithiasis. Upon admission, elevated liver enzymes (ASAT—432, ALAT—604, GGT—187) and bilirubin (131.1—total; 68.9—conjugated) were established. Leukocyte count and CRP were within normal range. ERCP revealed type I MS with an 8 mm stone impacted in the CD. CBD was normal and the common hepatic duct was dilated to 12 mm. Cannulation was achieved after "pre cut" sphincterotomy and a 10 fr/8 cm plastic stent was inserted to restore biliary drainage. There were no complications and the patient was scheduled for surgical treatment. LC was performed 2 weeks later, but due to severe inflammation in the Calot triangle and the presence of anatomical aberrations of the biliary tree, conversion to open cholecystectomy was decided. Despite this conversion, it was impossible to execute safe dissection, so a subtotal retrograde cholecystectomy was performed and the impacted stone was left in place. Therapeutic options including CA-LL were discussed with parents, but they declined further interventional procedures. Four months later, the patient was admitted with a new episode of jaundice, epigastric pain, nausea and vomiting, but no fever. Ultrasonography showed dilated intra- and extrahepatic bile ducts and obstruction of the stent was suspected. A new ERCP with stent exchange was performed. Rapid resolution of symptoms followed and the patient was discharged 2 days later. One month later, parents eventually consented for CA-LL. The procedure revealed a 15 mm stone impacted in the CD without fistulization, without strictures, and with only mild inflammatory changes in the CBD attributed to stenting. LL was performed, with consequent extraction of the fragments with a balloon. Flushing with saline followed to evacuate the sludge and debris from the bile ducts. The total duration was 40 min. The post-procedure period was uneventful and the patient was discharged on PP day 1. A total of three ERCP and one CA-LL session over 6 months were needed to achieve CD clearance. The patient has been symptom free so far (13 months).

Patients' information is summarized in Table 1.

**Table 1.** Summary of patient data.

| Patient | Age | Sex | Type of MS | Diagnostc Method | Number of ERCP/CA-LL Sessions | Duration of CA-LL Procedure (min) | Technical/Clinical Success | Adverse Events | Follow-Up (Months) |
|---|---|---|---|---|---|---|---|---|---|
| 1 | 66 | female | 1 | ERCP | 4/2 | 270 (180 + 90) | Yes/Yes | No | 13 |
| 2 | 44 | male | 1 | ERCP | 3/1 | 84 | Yes/Yes | Mild cholangitis | 17 |
| 3 | 37 | female | 1 | ERCP | 3/1 | 40 | Yes/Yes | No | 13 |

## 4. Discussion

Mirizzi syndrome is a relatively rare complication of gallstone disease. It has been reported in 0.7–25% of patients undergoing cholecystectomy [5]. First described more than a century ago, MS is still one of the most challenging diseases in the spectrum of biliary pathology. Though in general, surgery is considered the standard therapeutic approach, suboptimal clinical outcomes (15–20% morbidity), high rates of adverse events and high conversion rates (40%) for the laparoscopic modality, determined the need of new treatment options [7,15].

In the current literature there is already hard evidence supporting CA lithotripsy (either LL or EHL) as a rescue technique in difficult CBD stones. A recent meta-analysis by 24 Jin et al. evaluated the efficacy and safety of SOC-guided lithotripsy in treating difficult choledocholithiasis. The rate of complete stone clearance was 94% (95% CI, 90.2% to 97.5%). The rate of single-session stone clearance was 71.1% (95% CI, 62.1% to 79.5%) in the pooled 2786 patients. The number of sessions needed for complete stone clearance was 1.26 (95% CI, 1.17% to 1.34%) [23]. LL in particular is established to be highly effective with a ductal clearance rate of around 86–100% [24–26].

Endoscopic management of MS was first described 30 years ago, when Binmoeller et al. presented 14 cases of MS treated by peroral cholangioscopy combined with EHL.

Successful clearance of the CD was achieved in all patients (in 12 of them in single session) and only one complication was observed (bile leakage managed conservatively). Since then, there is accumulating evidence, mostly in the form of case reports or small case series, supporting the applicability and safety of ERCP not only for diagnosis, but for definitive treatment of MS [18,27,28]. Endoscopic management of MS could be performed in a conventional manner using fluoroscopic guidance and stone extraction using an extraction balloon or Dormia basket. Unfortunately, cannulation and advancement of accessories in the CD, especially in the settings of an impacted stone, is cumbersome and prone to failure. Currently the emphasis in the endoscopic management of MS is set on cholangioscopy-guided techniques and includes EHL or LL. Existing research show high success rate of cholangioscopy-assisted lithotripsy methods varying between 96–100% [28,29]. Only Sepe et al. report suboptimal performance of CA techniques with only 77% ductal clearance [30]. Of note, their results are based on 13 patients, which is quite insufficient to derive a solid conclusion. Research of Issa et al. and Shim et al. [31] show that CA-assisted techniques are superior to ESWL both in terms of success rate (81% for ESWL) as well as duration and difficulty of the procedure.

Even scarcer is the information on the endoscopic management of pMS. Surgical management in those patients is expected to be even more problematic having in mind the inflammatory changes in the Calot triangle and the additional adhesions consequent to prior surgery. This statement is supported by several studies. In 2007, Walsh et al. describes five pMS patients treated surgically, of whom only one was successfully managed laparoscopically [32]. In 2009, Palanivelu et al. presented a retrospective cohort of 15 patients with CDS managed laparoscopically. An average operating time of 103.4 min, hospital length of stay of 4 to 12 days and 13.33% morbidity were established; those results are considered to be quite disappointing [33]. Lastly, in a study by Kar et al. of 12 patients with pMS subjected to surgery, only seven could be treated laparoscopically, of whom five were eventually converted to open surgery [34]. These studies demonstrate the lack of a viable minimally invasive surgical alternatives for pMS.

Endoscopic management of pMS utilizing conventional techniques and fluoroscopic guidance is unlikely to be successful, since advancement of a balloon above the stone or engagement of the calculus with a basket are virtually impossible. Moon et al. reported a study on 19 patients, in whom fluoroscopy-guided EHL on CDS was performed [35]. Though there were no cases of bile leakage there were two cases of hemobilia. The safety of such approach therefore is to be determined. The remaining cases in the literature of pMS are managed almost exclusively through cholangioscopy combined with either EHL or LL.

In our article, we report management of pMS by means of CA-LL. In all three patients, correct diagnosis of MS type I was set prior to surgery through ERCP. Despite that, surgical treatment was unsuccessful and failed to achieve ductal clearance. This fact contradicts to the findings in the literature stating that preoperative diagnosis is the main determinant for treatment success during surgery. Those results might be explained by the severity of the inflammatory process, but also underline the importance of performing IOC to adequately delineate biliary anatomy, particularly in such unfavorable circumstances. In all of the reviewed cases, IOC was not performed.

In every patient, at least one attempt to remove the stone using conventional endoscopic techniques was made. In all three patients, entrapment of the stone with a basket or balloon extraction failed, which was largely anticipated.

In the current series we used the SpyGlass DS system (Boston Scientific, Marlborough, MA, USA) as the cholangioscopic device. Virtually all research on the subject nowadays utilizes the same system, which ensures good comparability of the results. The Holmium laser device set at 1.2 J/10 Hz combined with a 250 μm laser fiber was chosen for lithotripsy. These settings are comparable to the ones used in similar research [36]. A larger 500 μm laser fiber was selected in one case but was found to be too rigid, significantly reducing the maneuverability of the cholangioscope and eventually was replaced with the 250 μm

fiber. The optimal size of the laser fiber is yet to be determined, but our series suggests that 250 μm is possibly the optimal choice.

Ductal clearance was achieved in 100% of the patients using CA-LL. Those results are in concordance to the ones published in the literature and outline the excellent performance of the method for this specific indication [20,28,29]. In two patients, ductal clearance was achieved in single session. In one patient who had a large 4 cm stone impacted in the cystic duct, two sessions were required. Whether the size and number of stones is a determinant factor for successful ductal clearance is still unknown and urges further investigation. The presence of stricture in the cystic duct orifice is considered to be a predictor for therapeutic failure according to one study [28]. There were no strictures present in any of our cases.

The mean procedure duration was 98.5 min (40–180 min), which is similar to the existing data [30]. Close association between stone size and number and procedure time may be delineated and requires further study. One adverse event Grade I (Adverse events with any deviation of the standard postprocedural course, without the need for pharmacologic treatment or endoscopic, radiologic or surgical interventions) was established. It was in the form of mild cholangitis, which resolved within one day and required only usage of antipyretics. It is a well-known fact that cholangitis is the most common complication of cholangioscopy irrespective of indication (4%), and our series supports that statement.

Prophylactic stenting after laser lithotripsy is a matter of debate, but was not used in our series. Overall, in case ductal clearance is verified endoscopically and fluoroscopically and adequate drainage is achieved, it is our opinion that placement of a stent is not necessary and has no reflection on the clinical outcome.

In terms of cost, we established a mean value of CA-LL of EUR 1610.76 per procedure. Those expenses include only single-use consumables—cholangioscope, guidewires, sphincterotome, extraction balloon, Dormia basket, etc. The structure of the health system in Bulgaria precludes the precise estimation of general costs such as hospital staff and stay. Compared to a single conventional ERCP procedure (cost of around EUR 600), CA-LL is certainly more expensive. Taking into consideration the fact though, that "difficult" choledocholithiasis (including CDS) commonly requires three or four standard ERCP sessions, we would suggest that peroral cholangioscopy is cost-effective, particularly for this indication.

The current series has some strengths and limitations. Obvious limitations are the small case sample and retrospective selection. On the other hand, it reviews consecutive patients, which reduces selection bias. All procedures were performed by single endoscopist, which might suggest the importance of the learning curve and improvement of technique for the clinical outcome.

## 5. Conclusions

Our results demonstrate that CA-LL is a safe and effective treatment for pMS. Given the minimally invasive nature of the procedure, its high success rate and good safety profile, we suggest that patients with pMS should be routinely evaluated for endoscopic treatment. Further studies should be performed to evaluate the cost effectiveness of cholangioscopy for CDS and to directly compare a surgical cohort to a CA-LL cohort.

**Author Contributions:** Conceptualization, B.H. and D.R.; methodology, V.A. and E.T.; investigation, B.H. and D.R.; resources, D.D. and K.D.; data curation, K.K.; writing—original draft preparation, B.H.; writing—review and editing, S.V.; visualization, G.K. and P.U.; supervision, K.D. and S.V.; project administration V.A. and E.T.; funding acquisition, D.D. All authors have read and agreed to the published version of the manuscript.

**Funding:** This research received no external funding.

**Institutional Review Board Statement:** The study was conducted in accordance with the Declaration of Helsinki, and approved by the Ethics Committee of University hospital "Kaspela" (47/17.11.2022) for studies involving humans.

**Informed Consent Statement:** Informed consent was obtained from all subjects involved in the study. Written informed consent has been obtained from the patient(s) to publish this paper.

**Data Availability Statement:** Data available on request due to privacy restrictions.

**Conflicts of Interest:** The authors declare no conflict of interest.

## Abbreviations

| | |
|---|---|
| ALAT | alanine aminotransferase |
| AP | alkaline phosphatase |
| ASAT | aspartate aminotransferase |
| CA | cholangioscopy-assisted |
| CA-EHL | cholangioscopy-assisted electrohydraulic lithotripsy |
| CA-LL | cholangioscopy-assisted laser lithotripsy |
| CBD | common bile duct |
| CD | cystic duct |
| CHD | common hepatic duct |
| CDS | cystic duct stone |
| CT | computer tomography |
| EHL | electrohydraulic lithotripsy |
| ERCP | endoscopic retrograde cholangiopancreatography |
| ES | endoscopic sphincterotomy |
| ESWL | extracorporeal shockwave lithotripsy |
| EUS | endoscopic ultrasonography |
| GGT | gamma-glutamyl transpeptidase |
| IOC | intraoperative cholangiography |
| LL | laser lithotripsy |
| LC | laparoscopic cholecystectomy |
| MRCP | magnetic resonance cholangiopancreatography |
| MS | Mirizzi syndrome |
| pMS | postcholecystectomy Mirrizi syndrome |
| SOC | single-operator cholangioscopy |
| US | ultrasonography |

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
