# Peer review of "Cholangioscopy-Assisted Laser Lithotripsy for Treatment of Postcholecystectomy Mirizzi Syndrome: Case Series"

_gastroent, doi:10.3390/gastroent14010009_

Round 1
Reviewer 1 Report
#Cholangioscopy-assisted laser lithotripsy for treatment of postcholecystectomy Mirizzi syndrome: Case report#
The authors reported their experience with cholangioscopy-assisted laser lithotripsy for treatment of postcholecystectomy Mirizzi syndrome. The paper is clear, easy to follow and understand. However, the manuscript is well written, but there are some important methodological shortcomings in the review. My comments are the following:
- The manuscript reports a little case series of patients who received the same treatment that is the object of the study. I suggest to modify the title to" Cholangioscopy-assisted laser lithotripsy for treatment of postcholecystectomy Mirizzi syndrome: Case series"
- The introduction is too long, seems to be a discussion. Please revise it and summarize the focus of the study.
- In the section of the case reports patient 1 underwent a " retrograde" laparoscopic cholecystectomy and patient 2 underwent a" conventional antegrade" laparoscopic cholecystectomy. The conventional laparoscopic cholecystectomy is a retrograde one. Please check and eventually rephrase the sentences.
- I suggest to add some information about the cost of standard CA-LL procedure in the Discussion
Reviewer 2 Report
The authors presented a case series of 3 patients who underwent successful cholangioscopy assisted laser lithotripsy for management of postcholecystectomy Mirizzi Syndrome. The clinical and technical aspects of the cases were presented in a clear manner and are of educational value.
Minor comments:
1. In the conclusion section, the authors mentioned "Given the mini-invasiveness of the procedure, its high success rate and good safety profile, we suggest that patients with MS and particularly pMS should be routinely evaluated for 349 endoscopic treatment. ". Since this paper aims to report the role of endoscopy treatment for patients with postcholecystectomy Mirizzi Syndrome, perhaps authors can leave out "MS" and just focus on "pMS" in the conclusion.
2. English language / style editing of certain parts of the paper would be helpful (eg, in the above sentence, "Given the mini-invasiveness of the procedure......", the sentence may be rephrased as "Given the minimally invasive nature of the procedure...."
Reviewer 3 Report
Dear authors
1. The classification of MS should be simplified and explained in the text
2. The role of MRI should be better clarified
3. Why MRI was not performed at the f up instead of US that is less sensitive?
4. Please define safety according the ASGE lexicon
5. This is a case report not a case series because the number of subject enrolled is < 4
6. Furthermore, in the conclusions, it should be emphasized that t CA-LL could be a safe and effective treatment for pMS. Given the mini-invasiveness of the procedure, its high success rate and good safety profile, we suggest that patients with MS and particularly pMS could be routinely evaluated for endoscopic treatment.
7. References should be widely enriched through the citation of real life data about the use of cholangioscopy in difficult biliary stones
Sincerely
Round 2
Reviewer 1 Report
- A case series or case series report tipically includes 2 to 10 cases. I suggest to modify the title to" Cholangioscopy-assisted laser lithotripsy for treatment of postcholecystectomy Mirizzi syndrome: Case series"
- The introduction is too long, seems to be a discussion and the reader could lose attention and interest during the reading. The reader who searches for your article should know the topic and an explanatory introduction should not be needed. This section should be shortened.
Reviewer 2 Report
Authors have addressed the comments in prior review.
Reviewer 3 Report
All The issues were satisfactorily addressed
congratulation!
sincerely
Round 3
Reviewer 1 Report
No further comments